# The structure of liquid carbon elucidated by in situ X-ray diffraction

D. Kraus[1,2 ✉], J. Rips[1], M. Schörner[1], M. G. Stevenson[1], J. Vorberger[2], D. Ranjan[1,2], J. Lütgert[1], B. Heuser[1], J. H. Eggert[3], H.-P. Liermann[4], I. I. Oleynik[5], S. Pandolfi[6], R. Redmer[1], A. Sollier[7,8], C. Strohm[4], T. J. Volz[3], B. Albertazzi[9], S. J. Ali[3], L. Antonelli[10], C. Bähtz[2], O. B. Ball[11], S. Banerjee[12], A. B. Belonoshko[13], C. A. Bolme[14], V. Bouffetier[15], R. Briggs[3], K. Buakor[15], T. Butcher[12], V. Cerantola[16], J. Chantel[17], A. L. Coleman[3], J. Collier[12], G. W. Collins[18,19,20], A. J. Comley[21], T. E. Cowan[2], G. Cristoforetti[22], H. Cynn[3], A. Descamps[23], A. Di Cicco[24], S. Di Dio Cafiso[2], F. Dorchies[25], M. J. Duff[11], A. Dwivedi[15], C. Edwards[12], D. Errandonea[26], S. Galitskiy[5], E. Galtier[27], H. Ginestet[17], L. Gizzi[22], A. Gleason[27], S. Göde[15], J. M. Gonzalez[5], M. G. Gorman[3,28], M. Harmand[29,30], N. J. Hartley[27], P. G. Heighway[31], C. Hernandez-Gomez[12], A. Higginbotham[10], H. Höppner[2], R. J. Husband[4], T. M. Hutchinson[3], H. Hwang[4,32], D. A. Keen[33], J. Kim[34], P. Koester[35], Z. Konôpková[15], A. Krygier[3], L. Labate[35], A. Laso Garcia[2], A. E. Lazicki[3], Y. Lee[36], P. Mason[12], M. Masruri[2], B. Massani[11], E. E. McBride[23], J. D. McHardy[11], D. McGonegle[21], C. McGuire[3], R. S. McWilliams[11], S. Merkel[17], G. Morard[37], B. Nagler[27], M. Nakatsutsumi[15], K. Nguyen-Cong[3], A.-M. Norton[10], N. Ozaki[38], C. Otzen[39], D. J. Peake[31], A. Pelka[2], K. A. Pereira[40], J. P. Phillips[12], C. Prescher[39], T. R. Preston[15], L. Randolph[15], A. Ravasio[9], D. Santamaria-Perez[26], D. J. Savage[14], M. Schölmerich[41], J.-P. Schwinkendorf[2], S. Singh[3], J. Smith[12], R. F. Smith[3], J. Spear[12], C. Spindloe[12], T.-A. Suer[18], M. Tang[4], M. Toncian[2], T. Toncian[2], S. J. Tracy[42], A. Trapananti[24], C. E. Vennari[3], T. Vinci[9], M. Tyldesley[12], S. C. Vogel[14], J. P. S. Walsh[40], J. S. Wark[31], J. T. Willman[5], L. Wollenweber[15], U. Zastrau[15], E. Brambrink[15], K. Appel[15] & M. I. McMahon[11]

Carbon has a central role in biology and organic chemistry, and its solid allotropes provide the basis of much of our modern technology[1]. However, the liquid form of carbon remains nearly uncharted[2], and the structure of liquid carbon and most of its physical properties are essentially unknown[3]. But liquid carbon is relevant for modelling planetary interiors[4,5] and the atmospheres of white dwarfs[6], as an intermediate state for the synthesis of advanced carbon materials[7,8], inertial confinement fusion implosions[9], hypervelocity impact events on carbon materials[10] and our general understanding of structured fluids at extreme conditions[11]. Here we present a precise structure measurement of liquid carbon at pressures of around 1 million atmospheres obtained by in situ X-ray diffraction at an X-ray free-electron laser. Our results show a complex fluid with transient bonding and approximately four nearest neighbours on average, in agreement with quantum molecular dynamics simulations. The obtained data substantiate the understanding of the liquid state of one of the most abundant elements in the universe and can test models of the melting line. The demonstrated experimental abilities open the path to performing similar studies of the structure of liquids composed of light elements at extreme conditions.

Liquid carbon is difficult to produce in the laboratory[3,12]. It requires temperatures exceeding 4,000 K and pressures of at least several megapascals. In nature, these conditions are present in the interior of large planets such as the ice giants of our solar system, Uranus and Neptune, in which liquid carbon may contribute to the unusual magnetic fields of these planets[4,13]. Moreover, the equation of state of carbon is of substantial importance to estimate the composition of exoplanets from their observed mass and radius, in particular, for the highly abundant class of sub-Neptunes[5]. For technology applications, liquid carbon is an important transient state for the synthesis of several advanced carbon materials, such as carbon nanotubes[7], nanodiamonds[8,14] and Q-carbon[15].

Liquid carbon may also be key for the synthesis of the BC-8 phase of carbon, which has been predicted for decades at pressures beyond diamond stability[16,17] but could not be realized experimentally so far despite extensive efforts[18]. At the same time, carbon is used in inertial confinement fusion experiments as an ablator material surrounding the deuterium–tritium fuel[19]. The experimental design that achieved an ignited fusion plasma at the National Ignition Facility[20] relies on high-density carbon (diamond) that is subjected to shock compression just above melting in the initial phase of the implosion. This initial compression step is crucial for the subsequent fusion yield[21], and a better microscopic understanding of liquid carbon under dynamic

compression will help to design more efficient implosions, especially as more amorphous forms of carbon are considered as future ablator materials[22].

Modelling the extreme conditions in which liquid carbon prevails is challenging. Planetary interior pressures of around 100 GPa and temperatures approaching 10,000 K result in energy densities that are of the order of those stored in carbon–carbon bonds. Transient chemical bonds persisting from the $sp^3$-bonded diamond lattice are still expected to shape the structure of liquid carbon at these conditions, leading to tetrahedral coordination with four nearest neighbours on average, which is in contrast to simple liquids with icosahedral coordination with up to 12 nearest neighbours[3]. This complexity inhibits simple approximations and leaves first-principles simulations, usually based on density functional theory with molecular dynamics (DFT-MD), as the sole reliable method to provide predictive abilities. However, even DFT-MD requires assumptions, such as the choice of the exchange-correlation potential, and its computational limitations in system size and simulation times may not capture all relevant physics processes. There have been large discrepancies in the predictions of the melting curve of carbon, including deviations up to a factor of 2 in melting temperatures and fundamental differences in the slope of the melting curve for the diamond phase[23]. Machine-learning potentials based on DFT try to circumvent the scaling limitations by vastly extending spatial and temporal scales[24]; however, effects not covered by the training data are not necessarily expected to be captured by the scaled simulations. Although modern DFT-MD methods now seem to converge on high-pressure equilibrium phase diagram calculations with smaller variations[25–27], these predictions remain to be tested experimentally.

Probing the structural properties of carbon and other low-$Z$ fluids in static high-pressure experiments, for example, using diamond anvil cells, is challenging, and analogues such as glasses are used instead[28]. The high temperatures required to preserve the liquid state for prolonged periods lead to disintegration of the high-pressure sample confinement. For X-ray probing, the small scattering power of light elements often hampers resolving liquid structures above the background from surrounding material. In turn, most experimental approaches to study liquid carbon have used dynamic techniques such as electrical or optical flash heating and shock compression[3]. However, as in situ X-ray probing is difficult in these experiments, most results can up to now provide only indirect evidence for the presence of liquid carbon, and a detailed structure measurement has not yet been achieved. Velocimetry measurements in shock compression experiments can provide hints of melting through small changes in the slope of the shock Hugoniot curve[29]. Using pyrometry on decaying shocks, a temperature plateau was associated with melting, and an anomalously high specific heat in the liquid phase was interpreted to suggest a reconfiguration of atomic packing, from a partially bonded complex fluid to an atomic fluid between 10,000 K and 60,000 K (ref. 30). Measurements of electrical resistivity[31] and optical reflectivity[32] provided information on the conductivity of liquid carbon before X-ray sources with sufficient flux became available to attempt measurements of structural properties. X-ray absorption spectroscopy at soft X-ray sources using femtosecond flash heating, in which the dynamics can be benchmarked by extreme ultraviolet reflectivity[33], provided evidence for π and σ bonds in liquid carbon and some indirect information on structure based on theoretical modelling[34,35]. However, owing to the short timescales of these studies, the investigated states had not reached thermal equilibrium, and the theoretical methods applied require experimental benchmarking. A direct X-ray scattering measurement of the atomic structure of liquid carbon on the nanosecond timescale realized by laser-driven shock compression could be achieved only at a few distinct points in $k$-space, which leaves substantial degrees of freedom for models[36,37]. Synchrotrons and hard X-ray free-electron lasers finally started to allow for the measurement of diffraction patterns from liquids in

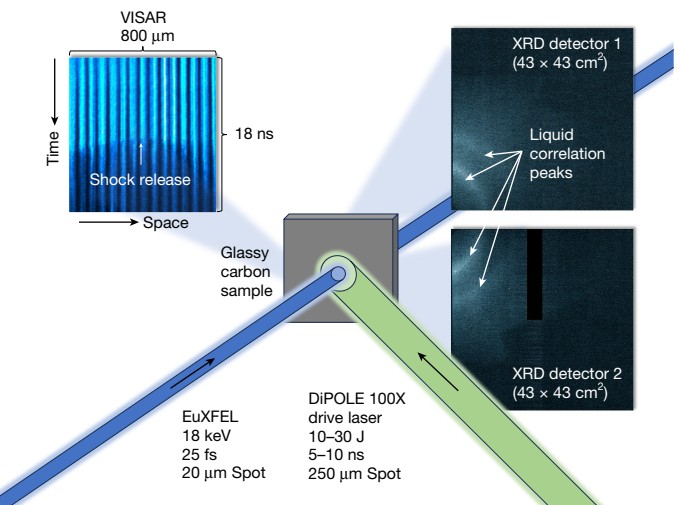

**Fig. 1 | Schematic of the experimental setup.** Glassy carbon samples were subjected to shock compression with the DiPOLE 100-X drive laser. The microscopic structure is probed by a bright X-ray pulse of EuXFEL, and two area detectors collect the XRD patterns. The shock dynamics are captured by a VISAR.

dynamic compression experiments reaching extreme conditions[38–42]. New facilities at the European XFEL (EuXFEL) set new standards in this direction[43].

The experiments reported here were performed using the DiPOLE 100-X high-energy laser at the High Energy Density-Helmholtz International Beamline for Extreme Fields (HED-HIBEF) instrument of EuXFEL[43,44]. Figure 1 shows the experimental setup. The DiPOLE 100-X laser was used to drive shock waves into glassy carbon samples, which generate high-pressure states with simultaneous heating due to the shock-induced entropy increase. Being a diode-pumped laser system, DiPOLE 100-X features an energy and temporal pulse shape stability at a sub-per-cent level, which allows for highly reproducible shot-to-shot drive conditions. The bright X-ray pulses delivered by EuXFEL with a photon energy of 18 keV were used for in situ X-ray diffraction (XRD) from the shock-compressed sample to monitor the microscopic structure. A velocity interferometer system for any reflector (VISAR) was used to capture the shock dynamics and determine the pressure, together with the diffraction data (Methods). Figure 2 shows integrated lineouts of single-shot diffraction patterns acquired before shock breakout time for increasing pressures. Ambient glassy carbon shows amorphous features associated with $sp^2$ bonds. At $(76 \pm 8)$ GPa, the $sp^2$ signature vanishes, and we observe a partial transformation of the glassy carbon to crystalline diamond. Remnants of the amorphous structure are still present at these conditions, possibly because of temperatures not considerably exceeding the glass transition temperature[45]. This changes at $(83 \pm 9)$ GPa, at which the diamond peaks substantially intensify and sharpen in comparison with the lower pressure conditions. This is consistent with the formation of larger crystallites and the probed sample volume being nearly fully composed of diamond. At $(106 \pm 11)$ GPa, the crystalline features start to diminish, together with the appearance of broader liquid correlation peaks. We interpret this as a coexistence state between diamond and liquid carbon; these features are also present at $(126 \pm 12)$ GPa but with lower diamond content. At $(160 \pm 14)$ GPa, we observe a purely liquid state.

From the diffraction patterns, we can extract the static structure factor profiles $S(k)$ for the covered scattering wavenumbers $k$, which allows for a direct comparison with ab initio simulations based on DFT-MD (Methods). In agreement with the simulations, we find a complex liquid with broad liquid correlation peaks forming around the positions of crystalline diamond (Fig. 3), which is compatible with transient bonds resulting in approximately fourfold coordination on average[27]. More

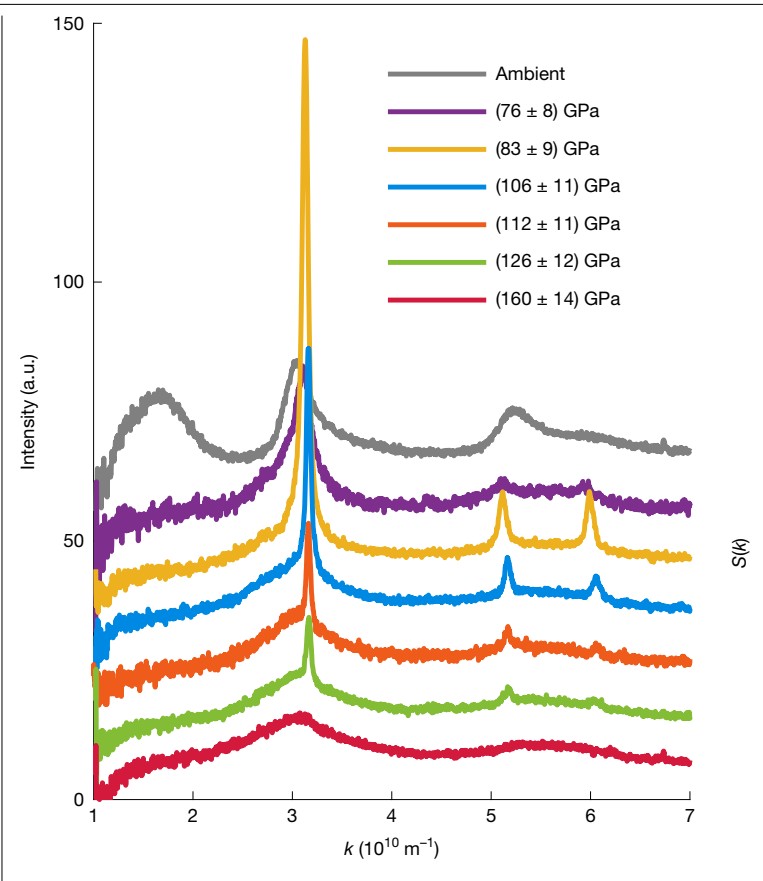

**Fig. 2 | XRD patterns of shock-compressed glassy carbon.** The drive pressure increases from top to bottom, and the curves are shown with a constant offset of 10 a.u. The X-ray probing times of the lineouts are within 1 ns before shock release to probe mostly homogeneous conditions and avoid pressure release states. At ambient conditions, broad amorphous structures are present. Diamond formation is observed above about 76 GPa, the coexistence of diamond and liquid carbon from about 100 GPa and complete melting at about 160 GPa. The quoted pressure uncertainties are dominated by the error estimations of the shock velocity measurement (Methods). a.u., arbitrary units.

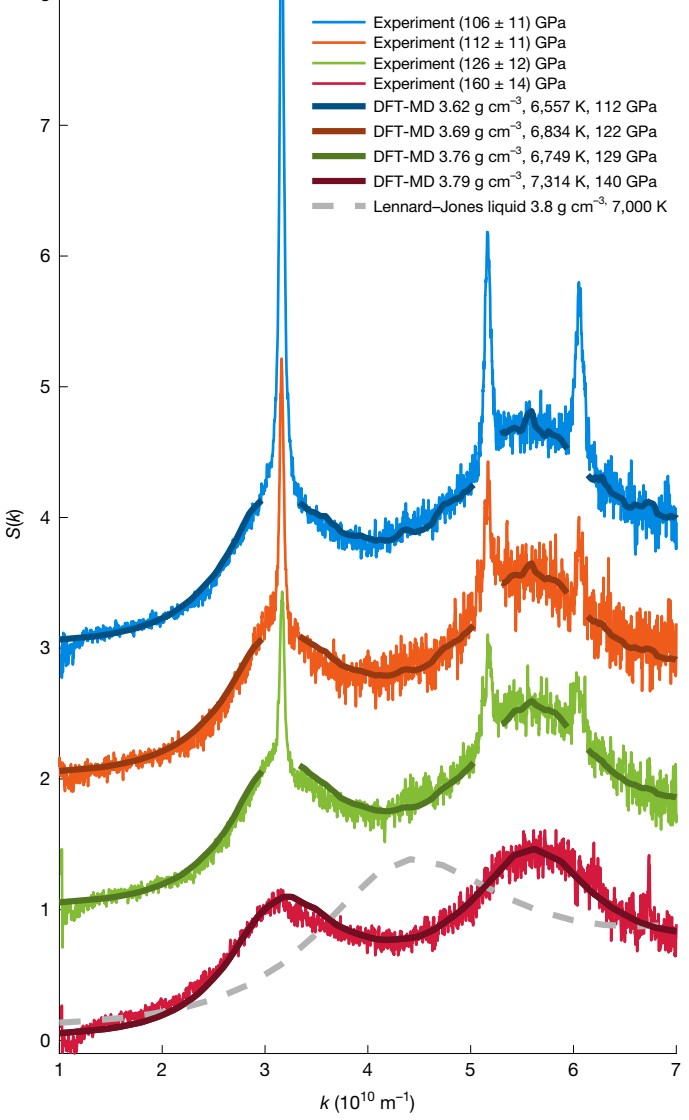

**Fig. 3 | DFT-MD fits of the experimentally obtained liquid structure.** The different datasets are shown with a constant offset of 1 between the curves. For the data with solid–liquid coexistence, the crystalline peaks have been omitted for fitting, and the liquid structure includes thermal diffuse scattering from diamond. The grey-dashed curve shows a hypernetted chain calculation of the liquid structure assuming a Lennard–Jones potential at similar conditions, which results in a first coordination number of around 11 and shows the fundamental difference to the observed structure, which has approximately fourfold coordination.

simplistic models with higher coordination numbers, such as Lennard–Jones, result in a first correlation peak between the two observed, and such a structure[37] is inconsistent with our observation. Calculating the Fourier transform of $S(k)$ allows us to determine the radial distribution function $g(r)$ and with that the first and second coordination numbers of the liquid carbon state. For the case of complete melting, Fig. 4a shows a range of experimental reconstructions with varying $k_{cutoff}$ (Methods). The span of results is in good agreement with the DFT-MD simulation that fits best to the corresponding XRD pattern. Again, a simple Lennard–Jones liquid does not match. Although the height and width of the first correlation peaks vary between the chosen cutoffs, the integrated area underneath the peaks and thus the extracted first and second coordination numbers remain rather constant. This is reasonable because the structural information is encoded in the XRD pattern, in which simple liquids with high coordination numbers are incompatible. For complete melting, we find a first coordination number of $3.78 \pm 0.15$ and a second coordination number of $17 \pm 2$, which is in agreement with several DFT-MD predictions of the bonded liquid[27,46–48] and our DFT-MD simulations (first coordination number of $3.66 \pm 0.05$).

By fitting with DFT-MD, we also infer estimates for the temperature and the density of the state reached within the probed volume. For the cases with solid–liquid coexistence, we fit a combination of liquid structure and thermal diffuse scattering of diamond, which can also be obtained from DFT (Extended Data Fig. 4). Theoretical predictions

expect the correlation peaks of $S(k)$ to move to higher $k$ with increasing density and broaden for higher temperatures[27] (Extended Data Fig. 3). Hence, we can provide experimental benchmarks for the melting temperature, the volume change from solid to liquid and the associated latent heat through the liquid structure at melting. The pressures extracted from the DFT-MD fits in Fig. 3 match with those obtained experimentally from VISAR and XRD reasonably well within the measurement uncertainty. Only for the highest pressure case, there is a small discrepancy, but still within the margins, because the uncertainties in temperature ($\pm 200$ K) and density ($\pm 0.05$ g cm$^{-3}$) of the DFT-fit result in a pressure error of around 8 GPa from the simulations. In the following, we use the pressures inferred from VISAR and the density from XRD. At ($106 \pm 11$) GPa, the positions of the crystalline diamond peaks result in a density of 3.91 g cm$^{-3}$, whereas the density of the liquid is best matched by 3.62 g cm$^{-3}$ with a temperature

**a**

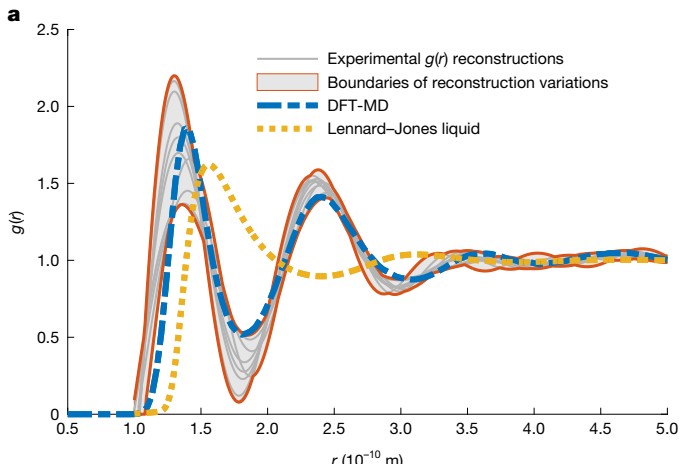
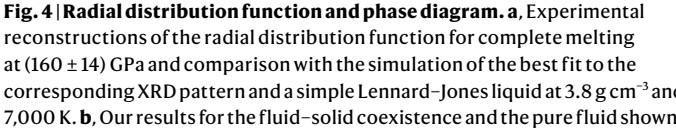

**b**

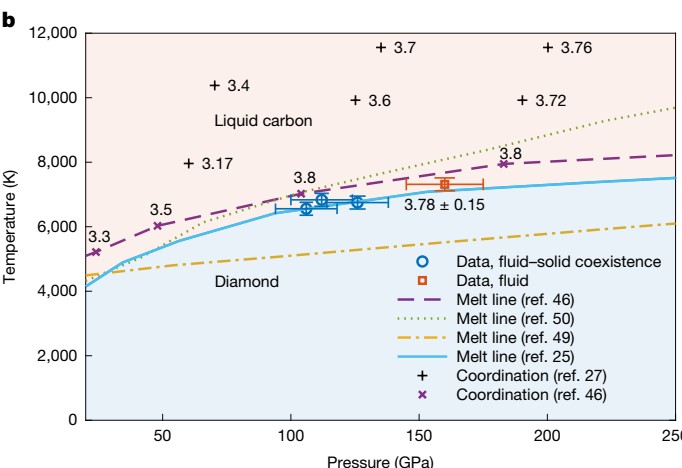

**Fig. 4 | Radial distribution function and phase diagram. a**, Experimental reconstructions of the radial distribution function for complete melting at (160 ± 14) GPa and comparison with the simulation of the best fit to the corresponding XRD pattern and a simple Lennard–Jones liquid at 3.8 g cm⁻³ and 7,000 K. **b**, Our results for the fluid–solid coexistence and the pure fluid shown in context with the predictions of the carbon melting line and coordination numbers for liquid carbon. The temperature uncertainties shown arise from the error estimations of the DFT-MD fits, and the pressure uncertainties correspond to the values quoted in Fig. 2.

of 6,557 K. The volume change of about (7 ± 1)% between the two phases is in reasonable agreement with DFT-MD predictions along the melting curve in this pressure regime (8% at 104 GPa and 6% at 183 GPa; ref. 46). Our data are also compatible with the pressure–temperature slope in ref. 25 of about 11.2 K GPa⁻¹. Using this value together with the experimentally determined volume change in the Clausius–Clapeyron relation, the entropy of melting and the latent heat are estimated to be about 20 J mol⁻¹ K⁻¹ and about 130 kJ mol⁻¹, respectively. The purely fluid state at (160 ± 14) GPa is best matched by a density of 3.79 g cm⁻³ and a temperature of 7,314 K.

Figure 4b shows these findings in the phase diagram of carbon with different DFT-MD predictions of the melting curve[25,46] and first coordination numbers[27,46]. The temperature values of our results are inferred from the comparison of the experimental and simulated structure curves. Thus, we assume that the structural representation of liquid carbon in our simulations is correct, which is corroborated by the excellent agreement with our diffraction data (for example, in comparison with the simple Lennard–Jones liquid). Although our deduced temperatures are not free from assumptions, we determine the liquid structure at melting with high precision, which will be a valuable benchmark for all future simulations of the melting transition of carbon in this regime. In general, we find high consistency with the more recent DFT-MD equation-of-state calculations in ref. 25, whereas other simulations[46] would require higher temperatures for melting, which is inconsistent with our results. Several melting curves based on more approximate models[49,50] do not match our data. Our measurements are expected to achieve thermodynamic equilibrium at the highest temperatures, given the robust diamond formation before melting, the exceeding of the glass transition temperature for glassy carbon and the observed consistency with equilibrium melting models, with that in best agreement[25], also having similar simulation settings to those used in our DFT-MD calculations. Moreover, we show the predicted coordination numbers of distinct DFT-MD simulations in refs. 27,46. Again, we find very good agreement with our measurements. The higher values predicted in ref. 46 match slightly better, but ref. 27 is also compatible within the experimental and numerical uncertainties.

In conclusion, our pioneering experiments substantiate the view of liquid carbon as a complex liquid with approximately fourfold coordination at pressures of about 100 GPa. Overall, the structures predicted by modern DFT-MD simulations are in agreement with the experimental data, which underlines the predictive power of this method at pressures around 100 GPa and elevated temperatures. It should be noted that all curves shown here were collected as single-shot events with a repetition rate in the range of minutes. However, both the drive laser and the X-ray probe can run at 10 Hz. Thus, future experiments can obtain higher precision by the accumulation of data and determine the liquid structure of a plethora of compounds made out of light elements at extreme pressure and temperature conditions. This could lead to substantial progress in models for the interior and the evolution of icy giant planets, the classification of exoplanets made of similar constituents, defining new processes of materials synthesis through extreme conditions, and designing the best ablator materials for inertial confinement fusion.

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

[1]Institut für Physik, Universität Rostock, Rostock, Germany. [2]Helmholtz-Zentrum Dresden-Rossendorf (HZDR), Dresden, Germany. [3]Lawrence Livermore National Laboratory, Livermore, CA, USA. [4]Deutsches Elektronen-Synchrotron DESY, Hamburg, Germany. [5]Department of Physics, University of South Florida, Tampa, FL, USA. [6]Sorbonne Université, Muséum National d'Histoire Naturelle, Insitut de Minéralogie, de Physique des Matériaux et de Cosmochimie (IMPMC), Paris, France. [7]CEA DAM Île-de-France, Arpajon, France. [8]Laboratoire Matière en Conditions Extrêmes, Université Paris-Saclay, CEA, Bruyères-le-Châtel, France. [9]Laboratoire pour l'utilisation des lasers intenses (LULI), Ecole Polytechnique, Palaiseau, France. [10]York Plasma Institute, School of Physics, Engineering and Technology, University of York, Heslington, UK. [11]SUPA, School of Physics and Astronomy, and Centre for Science at Extreme Conditions, The University of Edinburgh, Edinburgh, UK. [12]Central Laser Facility (CLF), STFC Rutherford Appleton Laboratory, Didcot, UK. [13]Frontiers Science Center for Critical Earth Material Cycling, School of Earth Sciences and Engineering, Nanjing University, Nanjing, China. [14]Los Alamos National Laboratory, Los Alamos, NM, USA. [15]European XFEL, Schenefeld, Germany. [16]Dipartimento di Scienze dell'Ambiente e della Terra, Università degli Studi di Milano Bicocca, Milano, Italy. [17]Université de Lille, CNRS, INRAE, Centrale Lille, Lille, France. [18]Laboratory for Laser Energetics, University of Rochester, Rochester, NY, USA. [19]Department of Physics and Astronomy, University of Rochester, Rochester, NY, USA. [20]Department of Mechanical Engineering, University of Rochester, Rochester, NY, USA. [21]AWE, Aldermaston, Reading, UK. [22]Istituto Nazionale di Ottica, CNR - Consiglio Nazionale delle Ricerche, Pisa, Italy. [23]School of Mathematics and Physics, Queen's University Belfast, Belfast, UK. [24]School of Science and Technology, Physics Division, Università di Camerino, Camerino, Italy. [25]CELIA, Université de Bordeaux, CNRS, CEA, Talence, France. [26]Departamento de Fisica Aplicada, Universidad de Valencia, Valencia, Spain. [27]SLAC National Accelerator Laboratory, Menlo Park, CA, USA. [28]First Light Fusion, Oxford, UK. [29]Institut Minéralogie, de Physique des Matériaux et de Cosmochimie (IMPMC), Sorbonne Université UMR CNRS, Paris, France. [30]PIMM, Arts et Metiers Institute of Technology, CNRS, Cnam, HESAM University, Paris, France. [31]Department of Physics, Clarendon Laboratory, University of Oxford, Oxford, UK. [32]Department of Environment and Energy Engineering, Gwangju Institute of Science and Technology (GIST), Gwangju, Korea. [33]ISIS Facility, STFC Rutherford Appleton Laboratory, Didcot, UK. [34]Department of Physics, Hanyang University, Seoul, South Korea. [35]CNR - Consiglio Nazionale delle Ricerche, Istituto Nazionale di Ottica, (CNR - INO), Florence, Italy. [36]Department of Earth System Sciences, Yonsei University, Seoul, South Korea. [37]University of Grenoble Alpes, University of Savoie Mont Blanc, CNRS, IRD, University of Gustave Eiffel, ISTerre, Grenoble, France. [38]Graduate School of Engineering, University of Osaka, Suita, Osaka, Japan. [39]Institut für Geo- und Umweltnaturwissenschaften, Albert-Ludwigs-Universität Freiburg, Freiburg, Germany. [40]Department of Chemistry, University of Massachusetts Amherst, Amherst, MA, USA. [41]Paul Scherrer Institut, Villigen, Switzerland. [42]Earth and Planets Laboratory, Carnegie Science, Washington, DC, USA. ✉e-mail: dominik.kraus@uni-rostock.de

## Methods

### Laser-driven shock compression and experimental pressure determination

The samples were subjected to shock compression using frequency-doubled pulses of the DiPOLE 100-X laser (515 nm wavelength) and a phase plate that produces a smoothed laser spot of approximately 250 μm in diameter. XRD was recorded by two Varex 4343CT flat panel detectors. For more details on the experimental configuration and geometry, see ref. 43. Quasi-flattop pulses of 10 ns with 35 J maximum energy and 5 ns with 28 J maximum energy resulted in steady shock compression waves. The thickness of the samples was either 60 μm or 92 μm. For each drive condition, we obtained a time series with intervals between different X-ray probe timings of 0.5–1 ns, and the data used for XRD fitting were within the last nanosecond before the shock release for the 6–10 ns of total transit time. We chose the timing before release to avoid the large density–pressure gradients afterwards. Signal from any remaining cold material can be accounted for by subtracting the ambient pattern that is recorded for each sample before the laser shot, with a scaling of the probe timing relative to the recorded shock release. When diamond peaks are present, the density of the crystallites can be determined with high precision and shows constant density within the measurement uncertainty of 0.01 g cm$^{-3}$ as long as the shock propagates inside the sample (Extended Data Fig. 1). Thus, the assumption of a planar steady shock is reasonably justified (Extended Data Fig. 2). Once the shock releases on the sample rear side, the density of the diamond crystallites approaches ambient density after a few nanoseconds and reaches even lower values afterwards because of the residual high temperatures. The different time series do not show substantial effects of X-ray preheating, as the XRD features associated with the weak $sp^2$ bonds diminish proportionally to the shock distance travelled and are not markedly affected right after the impact of the drive laser. As glassy carbon is not transparent to optical light, the VISAR system can determine only shock transit times, and the assumption of a steady shock propagation provides the shock velocity. In situ XRD allows for determining density as long as diamond and/or liquid carbon are present. With the obtained density and shock velocity, the shock pressure $P_S$ can be determined by the Rankine–Hugoniot relations[51]:

$$P_S = \rho_0 \left(1 - \frac{\rho_0}{\rho_S}\right) v_S^2. \tag{1}$$

The uncertainties of the resulting pressures are dominated by the uncertainty of the shock velocity due to the quadratic scaling. For the lower pressure conditions, there is very good overlap with gas gun Hugoniot measurements on glassy (vitreous) carbon up to 85 GPa (ref. 52; Extended Data Table 1 and Extended Data Fig. 1). For higher pressures, there are no existing Hugoniot data in the literature.

### Liquid diffraction analysis

The total scattered X-ray intensity $I(k)$ in our XRD patterns, which are corrected for transmission through aluminium filters in front of the detectors, is given by

$$I(k) = \alpha(f^2(k)S(k) + I_{inc}(k)), \tag{2}$$

where $\alpha$ is a scaling factor from atomic units to measured counts on the detector. The static structure factor $S(k)$ was then determined by using the atomic form factors $f(k)$ and incoherent scattering functions $I_{inc}(k)$ for carbon as tabulated in ref. 53. The normalization of the experimental $S(k)$ curve was included into the fitting procedure to the DFT-MD simulations by minimizing the function

$$\left(\frac{I(k)}{\alpha} - p_1 I_{inc}(k) - p_2 f^2(k) S(k)^{DFT}_{\rho,T}\right)^2, \tag{3}$$

where the ratio of the scaling parameters for incoherent and coherent scattering, $p_1$ and $p_2$, respectively, resulted in a constant value for all analysed datasets as the incoherent scattering is not affected by structural changes. For fitting the DFT-MD simulations, we cut the measured XRD pattern at $k_{cutoff} = 7$ Å$^{-1}$, because at higher angles, the applied filter and geometry corrections for the detector sensitivity become more severe[43] and would increase the uncertainty of our analysis. The radial distribution function

$$g(r) = 1 + \frac{1}{2\pi^2 nr} \int_0^{k_{cut-off}} [S(k) - 1] k\sin(kr)\mathrm{d}k \tag{4}$$

with atomic number density $n$ was obtained from $S(k)$ by linear extrapolation of the experimentally obtained curve for (160 ± 14) GPa to $k = 0$. The coordination numbers

$$N_C = 4\pi n \int_{r_1}^{r_2} r^2 g(r)\mathrm{d}r \tag{5}$$

were then obtained by integrating between the corresponding minima of $g(r)$ at $r_1$ and $r_2$. For the first coordination number, $r_1 = 0$ and $r_2$ is given by the first minimum of $g(r)$. For the second coordination number, $r_1$ is the first minimum of $g(r)$ and $r_2$ is the second. The provided error estimations of the coordination numbers were determined by reasonable variations of $k_{cutoff}$ between 6.8 Å$^{-1}$ and 7.5 Å$^{-1}$ for calculating $g(r)$ and the density uncertainty.

### DFT-MD simulations

All DFT-MD simulations in this work were performed with the Vienna ab initio simulation package (VASP)[54–56]. The electronic and ionic parts were decoupled by the Born–Oppenheimer approximation and, for fixed ion positions, the electronic problem was solved in the finite temperature DFT approach[57] using a projector-augmented wave pseudopotential (labelled PAW_PBE C_h)[58,59] and the Perdew–Burke–Ernzerhof functional[60] for the exchange-correlation contribution. A 2 × 2 × 2 Monkhorst–Pack sampling[61] was used for the k-space of a simulation box with 64 atoms, and a plane wave cutoff energy of 1,000 eV was used, in which the resulting structure curves show excellent agreement with calculations using 216 atoms, which have been performed for selected parameters. The molecular dynamics time step was $t = 0.2$ fs. A Nosé–Hoover thermostat was used with the Nosé mass set to 0.5 am, corresponding to 67 time steps (0.37 × 10$^{15}$ Hz). The number of the considered MD steps (for the calculation of the structure factors) is generally in the range of 10,000. To perform the least-square fitting to the diffraction data, the static ion–ion structure factor was computed on a density and temperature grid ranging from 3.6 g cm$^{-3}$ to 3.9 g cm$^{-3}$ and from 6,000 K to 8,000 K with four and five density and temperature increments, respectively. The simulations provide the three-dimensional particle density distribution $n(\mathbf{r}, t)$, and by Fourier transform and averaging, the structure factor $S(k)$ can be obtained[37]. Close to the melting line, finite size effects that can seed crystallization features were circumvented by training a high-dimensional neural network potential. The forces and energies predicted by DFT were learnt by a Behler–Parrinello high-dimensional neural network potential[62] implemented in the n2p2 software package[63,64]. For more details on this method, see ref. 65. The temperature control in all molecular dynamics simulations was performed by a Nosé–Hoover thermostat[66,67]. A finer resolution of static structure factors in the density–temperature plane was achieved by computing a neural-network-based three-dimensional representation of $S(k)$ in the $k$–$\rho$–$T$ space as suggested in ref. 68. The neural network is a feedforward neural network with linear connections

and layers with 3, 64, 1,024, 1,024, 1,024, and 1 neurons and ReLU activation between the layers. A benchmark of the interpolation grid result at 3.7 g cm$^{-3}$ and 8,000 K with a DFT-MD simulation using a box with 216 atoms at these conditions is shown in Extended Data Fig. 5.

## Data availability

Data recorded for the experiment at the European XFEL will be openly available at https://doi.org/10.22003/XFEL.EU-DATA-002740-00 once the data embargo of the experiment campaign 2740 has been lifted (17 May 2026). The corresponding run numbers are provided in the Extended Data Table 1. Before the end of the data embargo, all relevant data are available from the authors upon reasonable request.

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

**Acknowledgements** We acknowledge the European XFEL in Schenefeld, Germany, for the provision of X-ray free-electron laser beam time at the Scientific Instrument High Energy Density Science and would like to thank the staff for their assistance. We are indebted to the Helmholtz International Beamline for Extreme Fields user consortium for the provision of instrumentation and staff that enabled this experiment. The data are available (10.22003/XFEL. EU-DATA-002740-00). We acknowledge DESY (Hamburg, Germany), a member of the Helmholtz Association HGF, for the provision of experimental facilities. Parts of this research were carried out at PETRA III (beamline P02.2). The work of D.K., D.R., J.R. and M.G.S. was supported by the Deutsche Forschungsgemeinschaft (DFG, German Research Foundation) project no. 505630685. J.L. was supported by GSI Helmholtzzentrum für Schwerionenforschung, Darmstadt, as part of the R&D project SI-URDK2224 with the University of Rostock. K.A., K.B., Z.K., H.-P.L. and R.R. thank the DFG for support within the Research Unit FOR 2440. Part of this work was performed under the auspices of the US Department of Energy by the Lawrence Livermore National Laboratory (LLNL) under contract no. DE-AC52-07NA27344 and was supported by the Laboratory Directed Research and Development Program at LLNL (project no. 21-ERD-032). Part of this work was performed under the auspices of the US Department of Energy through the Los Alamos National Laboratory, operated by Triad National Security, for the National Nuclear Security Administration (contract no. 89233218CNA000001). Research presented in this letter was supported by the Department of Energy, Laboratory Directed Research and Development program at Los Alamos National Laboratory under project no. 20190643DR and at the SLAC National Accelerator Laboratory, under contract no. DE-AC02-76SF00515. This work was supported by grant nos. EP/S022155/1 (M.I.M., M.J.D.) EP/S023585/1 (A.H., L.A.) and EP/S025065/1 (J.S.W.) from the UK Engineering and Physical Sciences Research Council. J.D.M. is grateful to AWE for the award of CASE Studentship P030463429; P.G.H. acknowledges support from OxCHEDS through AWE (PDRA contract no. 30469604). E.E.M. and A. Descamps were supported by the UK Research and Innovation Future Leaders Fellowship (MR/W008211/1) awarded to E.E.M.; D.E. and D.S.-P. from the University of Valencia thank the financial support by the Spanish Ministerio de Ciencia, Innovacion y Universidades and the Agencia Estatal de Investigacion (MCIN/AEI/10.13039/501100011033) under grant nos. PID2021-125518NB-I00 and PID2022-138076NB-C41 (cofinanced by EU FEDER funds), and by the Generalitat Valenciana under grant nos. CIPROM/2021/075, CIAICO/2021/241 and MFA/2022/007 (funded by Next Generation EU PRTR-C17.I1). N.J.H. and A.G. were supported by the DOE Office of Science, Fusion Energy Science under FWP 100182. This material is based on the work supported by the Department of Energy National Nuclear Security Administration under award no. DE-NA0003856 Y.L. is grateful for the support from the Leader Researcher program (NRF-2018R1A3B1052042) of the Korean Ministry of Science and ICT (MSIT). B.M. and R.S.M. acknowledge funding from the European Research Council (ERC) under the Horizon 2020 research and innovation programme of the European Union (grant agreement no. 101002868). G.W.C. and T.-A.S. acknowledge partial funding from the Department of Energy National Nuclear Security Administration under award no. DENA0003856 and the Center for Matter at Atomic Pressures, an NSF Physics Frontier Center, award no. PHY-2020249. S.M., H.G. and J. Chantel were funded by the European Union (ERC, HotCores, grant no. 101054994). Views and opinions expressed are, however, those of the author(s) only and do not necessarily reflect those of the European Union or the ERC. Neither the European Union nor the granting authority can be held responsible for them. S.P. acknowledges support from the GOtoXFEL 2023 AAP from CNRS and the ANR grant HEX-DYN (ANR-24-CE30-4792). N.O. was supported by grants from JSPS KAKENHI (grant no. 23K20038), JSPS Core-to-Core program (JPJSCCA20230003) and MEXT Q-LEAP (JP-MXS0118067246). The work of I.I.O., S. Galitskiy, J.M.G. and J.T.W. was supported by the Academic Collaborative Team award of the LLNL and DOE/NNSA (award nos. DE-NA-0003910 and DE-NA-0004089) and DOE/FES (award nos. DE-SC0023508 and DE-SC0024640).

**Author contributions** D.K., J.R., M. Schörner, M.G.S., J.V., D.R., J.L., B.H., J.H.E., H.P.L., I.I.O., S.P., R.R., A.S., C. Strohm, T.J.V., B.A., S.J.A., L.A., C.B., O.B.B., S.B., A.B.B., C.A.B., V.B., R.B., K.B., T.B., V.C., J. Chantel, A.L.C., J. Collier, G.W.C., A.J.C., T.E.C., G.C., H.C., A. Descamps, A.D.-C., S.D., F.D., M.J.D., A. Dwivedi, C.E., D.E., S. Galitskiy, E.G., H.G., L.G., A.G., S. Göde, J.M.G., M.G.G., M.H., N.J.H., P.G.H., C.H.-G., A.H., H. Höppner, R.J.H., T.M.H., H. Hwang, D.A.K., J.K., P.K., Z.K., A.K., L.L., A.L., A.E.L., Y.L., P.M., M.M., B.M., E.E.M., J.D.M., D.M., C.M., R.S.M., S.M., G.M., B.N., M.N., K.N.-C., A.-M.N., N.O., C.O., D.J.P., A.P., K.A.P., J.P.P., C.P., T.R.P., L.R., A.R., D.S.-P., D.J.S., M. Schölmerich, J.-P.S., S.S., J. Smith, R.F.S., J. Spear, C. Spindloe, T.A.S., M. Tang, M. Toncian, T.T., S.J.T., A.T., C.E.V., T.V., M. Tyldesley, S.C.V., J.P.S.W., J.S.W., J.T.W., L.W., E.B., U.Z., K.A. and M.I.M. were part of the collaboration that conceived, planned and performed the experiment. The DFT-MD simulations were performed and analysed by M. Schörner and J.V. The analysis of the experimental data was performed by J.R., M.G.S., D.R., J.L. and D.K. with discussion input from J.H.E., H.-P.L., I.I.O., S.P., R.R., A.S., C. Strohm and T.J.V. The paper was written by D.K. and reviewed by all authors.

**Funding** Open access funding provided by Universität Rostock.

**Competing interests** The authors declare no competing interests.

**Additional information**
**Correspondence and requests for materials** should be addressed to D. Kraus.

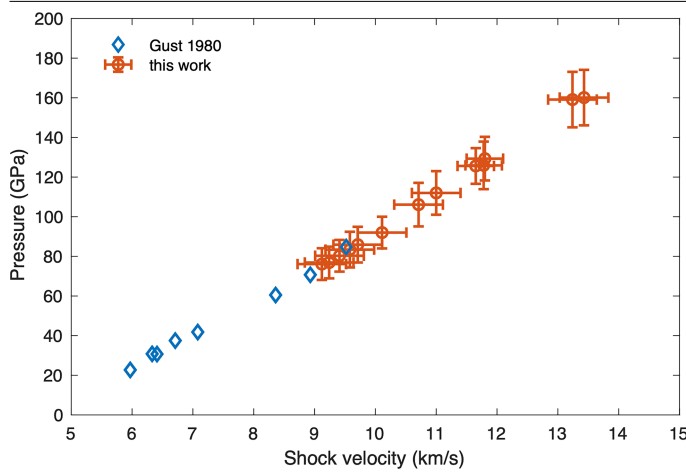

**Extended Data Fig. 1 | Glassy carbon Hugoniot data.** The inferred pressures for the obtained shock velocities connect very well to previous gas gun shock Hugoniot measurements of vitreous carbon by W. H. Gust (ref. 52 of the article).

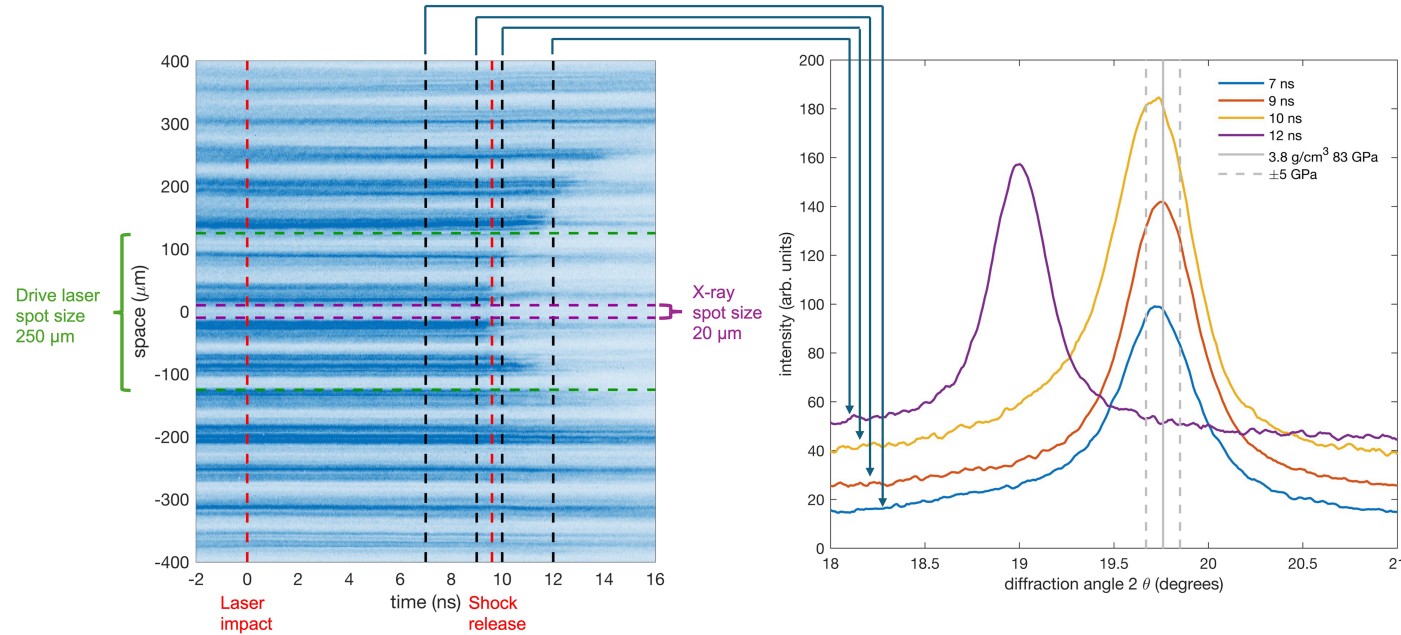

**Extended Data Fig. 2 | Assessment of the steadiness of the shock and potential influences of spatial gradients.** Left: Examplary line VISAR image showing the moment of shock release for different positions on the sample rear side (run 551). Since the X-rays probe the small central area of the laser-compressed sample, the shock transit time only varies by 0.3 ns within the region probed by the X-rays. For stronger drives and thinner samples, this variation is even smaller. The steadiness of the shock is underlined by taking X-ray diffraction snapshots at different time delays for the same drive conditions. While the shock is inside the sample, the pressure inferred from the diamond (111) XRD peak remains constant within 2 %. Even at 0.4 ns after the shock release, most of the sample remains at high pressure, while small portions of the XRD feature already shift to lower densities resulting in a slight asymmetry of the Bragg reflection. At 2.4 ns after the shock release, the whole sample is released to lower densities slightly below ambient density of diamond.

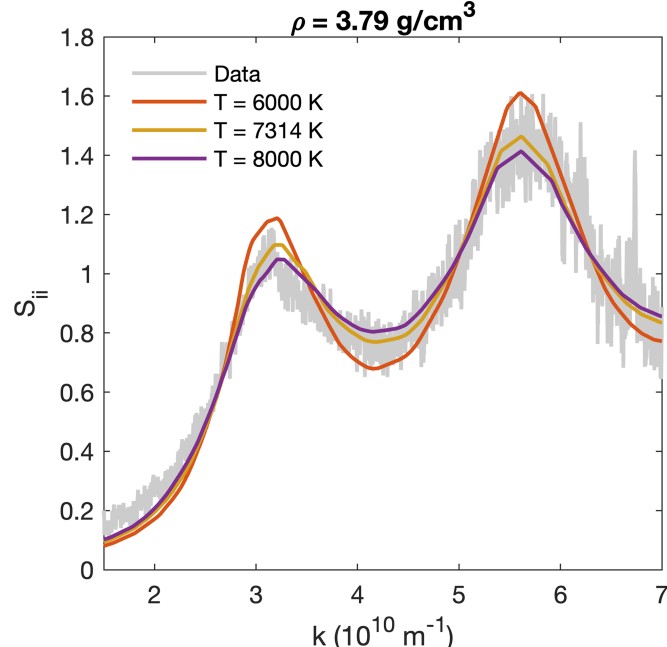
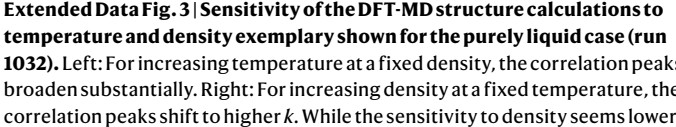
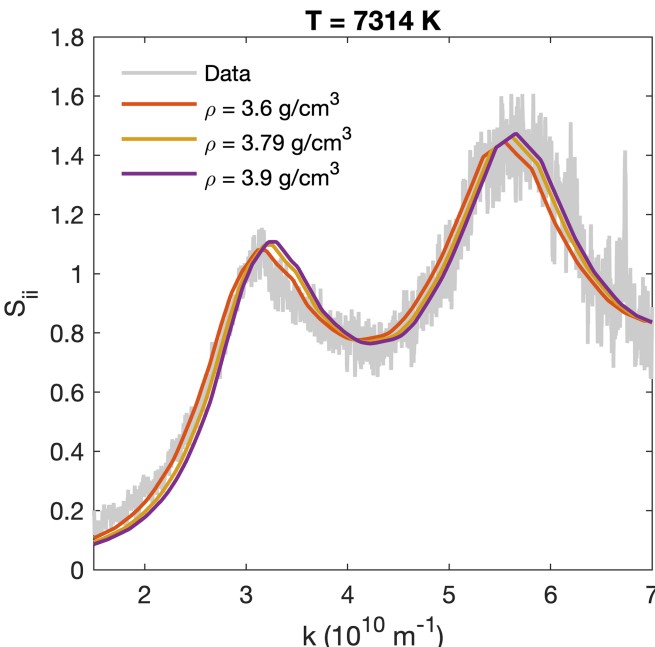

**Extended Data Fig. 3 | Sensitivity of the DFT-MD structure calculations to temperature and density exemplary shown for the purely liquid case (run 1032).** Left: For increasing temperature at a fixed density, the correlation peaks broaden substantially. Right: For increasing density at a fixed temperature, the correlation peaks shift to higher $k$. While the sensitivity to density seems lower than to temperature, it should be noted that the density is only varied by 8 % in total (in contrast to 29 % for the temperature), which is enough to provide stable fits to the obtained liquid structure data. The best fit parameters are $T = 7,173$ K and $\rho = 3.79$ g/cm$^3$ for the depicted case.

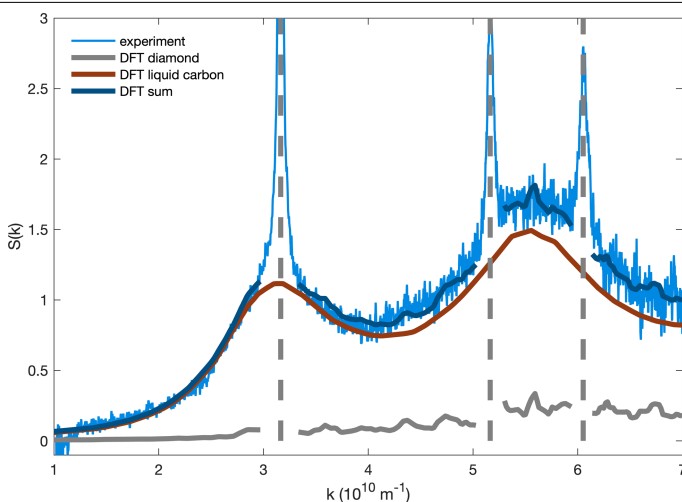

**Extended Data Fig. 4 | Fitting the liquid structure for the data showing diamond-liquid coexistence.** For the elevated temperatures of the experiments, there is also thermal diffuse scattering of crystalline diamond, which needs to be considered when fitting density and temperature to the liquid structure. This contribution was determined from DFT-MD simulations of diamond (see refs. 36 and 37 of the article). In the case shown as example (run 547), the diamond content is about 40 % which is obtained by scaling the amount of crystalline diffraction to the pure diamond case (e.g., run 551). While the thermal diffuse background is nearly linear with $k$ and given that the temperature is constrained by the broadening of the correlation peaks and the density by their position, the influence of the thermal diffuse scattering on the inferred temperature and density values is negligible. For the other coexistence cases shown in the article, the diamond content is lower: ~20 % for run 549 and ~10 % for run 1026.

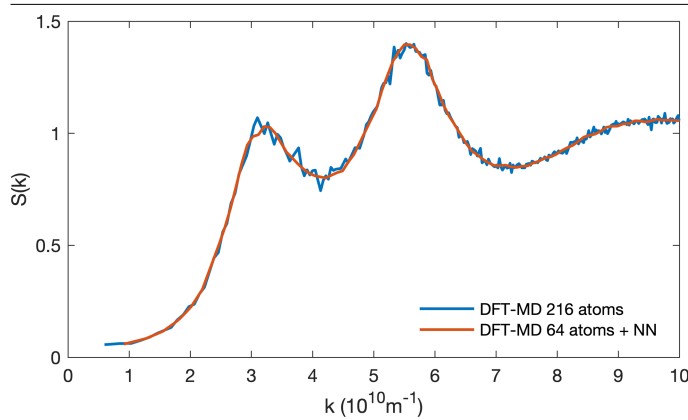

**Extended Data Fig. 5 | DFT-MD + NN upscaling and interpolation grid validation.** As an example, we depict a benchmark of $S(k)$ for 3.7 g/cm$^3$ and 8,000 K obtained from our DFT-MD interpolation grid based on a 64 atoms box and a neural network (NN) to a DFT-MD run using a larger box with 216 atoms. The two cases show excellent agreement.

**Extended Data Table 1 | Hugoniot data**

| run | $I_L$(TW/cm$^2$) | $d_s$(μm) | $t_t$(ns) | $v_s$ (km/s) | $\rho_d$(g/cm$^3$) | $\rho_l$(g/cm$^3$) | $P$(GPa) | $t_X$(ns) | state |
|---|---|---|---|---|---|---|---|---|---|
| 547 | 6.8 | 92 ± 1 | 8.6 ± 0.3 | 10.7 ± 0.4 | 3.91 ± 0.01 | 3.62 ± 0.05 | 106 ± 11 | 8.0 | dia + liq |
| 549 | 6.8 | 92 ± 1 | 8.4 ± 0.3 | 11.0 ± 0.4 | 3.91 ± 0.01 | 3.69 ± 0.05 | 112 ± 11 | 7.0 | dia + liq |
| 551 | 4.5 | 92 ± 1 | 9.6 ± 0.3 | 9.6 ± 0.4 | 3.80 ± 0.01 | - | 83 ± 9 | 9.0 | dia |
| 1013 | 3.75 | 60 ± 1 | 6.5 ± 0.2 | 9.2 ± 0.4 | 3.75 ± 0.01 | - | 76 ± 8 | 6.0 | dia |
| 1026 | 7.6 | 92 ± 1 | 7.6 ± 0.2 | 11.6 ± 0.3 | 3.92 ± 0.01 | 3.76 ± 0.05 | 126 ± 11 | 7.5 | dia + liq |
| 1032 | 11.3 | 92 ± 1 | 6.9 ± 0.2 | 13.3 ± 0.4 | - | 3.79 ± 0.05 | 160 ± 14 | 6.0 | liq |

Measured shock Hugoniot parameters used for the X-ray diffraction data shown in Figs. 2 and 3 of the article for different laser intensities $I_L$. The shock velocity $v_s$ was determined by dividing the measured sample thickness (micrometer measurement uncertainty of 1 μm) $d_s$ through the shock transit time $t_t$ assuming a steady shock. The density of crystalline diamond $\rho_d$ was determined via the lattice spacing inferred from the powder diffraction peaks. The density of the liquid state $\rho_l$ was obtained from the DFT-MD fits shown in Fig. 3 of the article. The pressure $P$ was calculated via the Hugnoniot relations and $t_X$ shows the X-ray probe time for the different experiments.