## [Peer Review File · Nature]

The structure of liquid carbon elucidated by in situ X-ray diffraction

Corresponding Author: Professor Dominik Kraus

Version 0:

Reviewer comments:

Referee #1

(Remarks to the Author)

THIS PAPER ADDRESSES AN IMPORTANT PROBLEM THAT IS IMPORTANT TO A WIDE SCIENTIFIC AUDIENCE. THE EXPERIMENTS REPORTED HERE ARE AT THE ABSOLUTE STATE OF THE ART, AND THE ANALYSIS IS APPROPRIATE, REVEALING A FLUID WITH FOUR NEAREST NEIGHBORS, IN AGREEMENT WITH THEORETICAL CALCULATIONS. THE PAPER CAN BE PUBLISHED "AS IS", BUT WITH ONE MINOR ADDITION. PLEASE ADD THIS REFERENCE TO VERY RECENT MEASUREMENTS OF OPTICAL REFLECTIVITY OF LASER MELTED CARBON IN THE INTRODUCTION:

Raj, S. L., Devlin, S. W., Mincigrucci, R., Schwartz, C. P., Principi, E., Bencivenga, F., Foglia, L., Gessini, A., Simoncig, A., Kurdi, G., Masciovecchio, C., Saykally, R. J. "Free Electron Laser Measurement of Liquid Carbon Reflectivity in the Extreme Ultraviolet ϵ ". *Photonics* 7(2), 35 (2020).

Referee #2

(Remarks to the Author)

The manuscript reports the results of measurements of the local (atomic) structure of a complex liquid (carbon) at extreme conditions of temperature and pressure. This is achieved by measuring the structure factor and systematically comparing measurements with DFT simulations. Measuring the structure factor of a liquid at melting and ~ 1 Mbar with such a low signal/noise ratio is by itself a major achievement and one that, in my opinion, warrants publication in a high-profile journal like Nature. In addition, measurements cover a sufficiently large momentum range to allow for the determination, by Fourier inversion, of the real-space pair distribution function. DFT simulations are used here primarily to extract the temperature of the samples, a quantity that is experimentally inaccessible. In my opinion the manuscript deserves to be published in Nature in some form. However I have a number of comments that I would like the authors to consider before the paper can be published. They are listed here in decreasing level of relevance:

1. A closer look at Fig. 2 shows that the amorphous/liquid component is present at all pressures, and not just above ~ 100 GPa, as stated in the caption. In the assumption of thermal equilibrium, the observation of coexistence between crystalline and disordered forms implies that the temperature coincides with T_m . Are the authors implying that measurements at 76 and 83 GPa are also at T_m ? And if they are not at T_m , why are these two points not at thermal equilibrium while the other points are? It is possible that the disordered component at 76 and 83 GPa is still below the glass transition (which is the reason why at $P=0$ the system is obviously not at equilibrium). However the observation raises a serious questions regarding thermal equilibrium (or lack thereof) in the experiments. In short, how do the authors know if/when thermal equilibrium is reached (including at the highest pressures)?

2. The use of a NN potential is crucial to obtain the results shown in Fig. 3 and for the temperature fitting (NB: DFT-MD with 64 atoms ($L=6\text{\AA}$) would lead to an uncertainty of about $2\pi/L \sim 1 \times 10^{-10} \text{ m}^{-1}$ in the determination of the structure factor purely from DFT simulations). The authors must provide more details (as supplementary information) regarding the construction and validation of the NN potential and why they believe it interpolates the full DFT $S(k)$ with the accuracy required to extract temperatures through fits shown in Fig. S3.

3. The use of DFT to determine temperatures is commendable (and DFT is certainly the highest level of theory that can be realistically used to do that). However, while the authors are adamant about the fact the uncertainties introduced by DFT can be controlled only partially, they are less explicit about the implications of such uncertainties on their main claims. For example, it is incorrect to refer to the data in Fig. 4b as "experimental data". The vertical axis is determined by simulations, not by experiments (as instead correctly stated in the main text). Similarly, at page 11, the authors say: "Thus, we assume that the structural representation of liquid carbon in our simulations is correct, which is corroborated by our data." I'm not sure what the authors mean by this sentence, as T_m is determined by DFT also in the other data reported in Fig. 4b, so Fig. 4b cannot be taken as a validation of the experiments.

4. The comparison with "simple" liquids is interesting but differences between carbon and a simple liquid are not as surprising as the authors claim. At page 6, for example, they state that coordination in carbon "is in contrast to most other liquids". I would rather say that it is in contrast to simple liquids. There are plenty of other "complex" liquids observed or predicted at those conditions.

Referee #3

(Remarks to the Author)

This manuscript presents novel results from laser-driven shock compression experiments of glassy carbon samples that were probed by an x-ray free electron laser (XFEL). The experimental methodology was consistent with previous work within the field. The bright XFEL x-rays were used for x-ray diffraction (XRD) measurements to provide structural information of the carbon samples. Complementary velocimetry measurements provided the pressures of the shocked states up to 160 GPa. The XRD data of the high-pressure liquid carbon was compared with density functional theory (DFT) coupled to molecular dynamics (MD) simulations. Overall, the experimental data was of good quality with appropriate uncertainties. Comparison of the data with the DFT-MD simulations were valid, which demonstrated the complex structure of high-pressure liquid carbon. The abstract, introduction, and conclusions were clear, and appropriate references were provided.

To improve the manuscript, the authors should address the following:

1. Figure 1. The VISAR image should show the scale lengths of time and space axes (e.g. # of ns, and # of μm). Similarly, the XRD detectors' images should show their sizes (e.g. # of mm). Also, the "liquid correlation peaks" are very faint, could the contrast of the images be improved?
2. Figure 2. The "blue" line colors for the ambient and 106 GPa states are very similar. Could another color be used for the ambient state (e.g. black)?
3. Methods. A brief description of the XRD detectors should be included.

Version 1:

Reviewer comments:

Referee #2

(Remarks to the Author)

The authors have addressed in a satisfactory manner all my comments. The manuscript can now be published.
--Sandro Scandolo

Referee #3

(Remarks to the Author)

The revised manuscript has addressed my previous points, and it could be published as is.

We thank the referees for taking the time to assess our manuscript and improve its quality. Please find our point-by-point response below:

Referee 1:

THIS PAPER ADRESSES AN IMPORTANT PROBLEM THAT IS IMPORTANT TO A WIDE SCIENTIFIC AUDIENCE. THE EXPERIMENTS REPORTED HERE ARE AT THE ABSOLUTE STATE OF THE ART, AND THE ANALYSIS IS APPROPRIATE, REVEALING A FLUID WITH FOUR NEAREST NEIGHBORS, IN AGREEMENT WITH THEORETICAL CALCULATIONS. THE PAPER CAN BE PUBLISHED "AS IS", BUT WITH ONE MINOR ADDITION. PLEASE ADD THIS REFERENCE TO VERY RECENT MEASUREMENTS OF OPTICAL REFLECTIVITY OF LASER MELTED CARBON IN THE INTRODUCTION:

Raj, S. L., Devlin, S. W., Mincigrucci, R., Schwartz, C. P., Principi, E., Bencivenga, F., Foglia, L., Gessini, A., Simoncig, A., Kurdi, G., Masciovecchio, C., Saykally, R. J. "Free Electron Laser Measurement of Liquid Carbon Reflectivity in the Extreme Ultraviolet e2. Photonics 7(2), 35 (2020).

We thank the referee for these supportive comments. The suggested reference has been added as No. 33 in the sentence beginning at line 188 of the manuscript): “X-ray absorption spectroscopy at soft X-ray sources using femtosecond flash-heating, where the dynamics can be benchmarked by extreme ultraviolet reflectivity³³, provided evidence for \$\pi\$ and \$\sigma\$ bonds in liquid carbon and some indirect information on structure based on theoretical modeling^{34,35}.”

Referee 2:

The manuscript reports the results of measurements of the local (atomic) structure of a complex liquid (carbon) at extreme conditions of temperature and pressure. This is achieved by measuring the structure factor and systematically comparing measurements with DFT simulations. Measuring the structure factor of a liquid at melting and ~ 1 Mbar with such a low signal/noise ratio is by itself a major achievement and one that, in my opinion, warrants publication in a high-profile journal like Nature. In addition, measurements cover a sufficiently large momentum range to allow for the determination, by Fourier inversion, of the real-space pair distribution function. DFT simulations are used here primarily to extract the temperature of the samples, a quantity that is experimentally inaccessible. In my opinion the manuscript deserves to be published in Nature in some form. However, I have a number of comments that I would like the authors to consider before the paper can be published. They are listed here in decreasing level of relevance:

We thank the referee for the supportive comments and the suggestions for improvement and clarification. We comply with all requests of the referee as described below:

1. A closer look at Fig. 2 shows that the amorphous/liquid component is present at all pressures, and not just above ~100 GPa, as stated in the caption. In the assumption of thermal equilibrium, the observation of coexistence between crystalline and disordered forms implies that the temperature coincides with T_m . Are the authors implying that measurements at 76 and 83 GPa are also at T_m ? And if they are not at T_m , why are these two points not at thermal equilibrium while the other points are? It is possible that the disordered component at 76 and 83 GPa is still below the glass transition (which is the reason why at $P=0$ the system is obviously not at equilibrium). However, the observation raises a serious questions regarding thermal equilibrium (or lack thereof) in the experiments. In short, how do the authors know if/when thermal equilibrium is reached (including at the highest pressures)?

We thank the referee for raising this issue, which requires discussion and clarification. The referee is correct that the initial sample, glassy carbon, is in a metastable state at ambient conditions. With increasing shock pressure, the temperature is increasing in our sample.

At 76 GPa, we find first clear evidence of diamond structures in the X-ray diffraction pattern. This pressure is substantially higher than the equilibrium pressure where diamond becomes the most stable form of carbon. The referee is correct that at these conditions, we are not in equilibrium, since the X-ray diffraction patterns clearly show remaining amorphous structures. Thus, the transition to diamond seems indeed not complete at this pressure, possibly because the temperature is below or at least not substantially above the glass transition temperature ($\sim 0.66 T_m$, see Ref. 45 of the revised manuscript). The kinetics of the shock induced transition of glassy carbon to diamond is an interesting topic on its own but not the topic of this paper (it may be covered in a follow-up publication after additional studies).

At 83 GPa, the sample is fully dominated by the diamond structure. The sharp diamond peaks are sitting on a small broader structure, which may result from defects or thin regions of higher temperatures close to the ablation plasma. However, the diamond structure vastly dominates, which is not compatible with coexistence of diamond and liquid carbon.

The situation changes at 106 GPa, where the liquid feature becomes clearly visible, and the crystalline diamond peak intensity decreases with increasing pressure from there on. This we interpret as solid-liquid coexistence up to complete melting observed at 160 GPa. We find that the temperatures for the coexistence cases extracted from the DFT-MD fits are consistent with the equilibrium melting line of Benedict et al. 2014, who used a very similar simulation setup as for our calculations. Therefore, we are confident that within the temperature error bars of our study, the assumption of thermal equilibrium is reasonable justified. The nanosecond timescale of our experiment is still long in comparison to femtosecond or picosecond timescales of DFT-MD simulations. To clarify these points in our manuscript, we made the following changes and additions:

- Description of experimental XRD patterns (starting at line 214 of the manuscript): “At (76±8) GPa, the sp² signature vanishes and we observe a partial transformation of the glassy carbon to crystalline diamond. Remnants of the amorphous structure are still present at these conditions, possibly due to temperatures not considerably exceeding the glass transition temperature⁴⁵. This changes at (83±9) GPa, where the diamond peaks substantially intensify and sharpen in comparison to the lower pressure conditions. This is consistent with the formation of larger crystallites and the probed sample volume being nearly fully composed of diamond. At (106±11) GPa, the crystalline features start to diminish together with the appearance of broader liquid correlation peaks. This we interpret as a coexistence state between diamond and liquid carbon; these features are also present at (126±12) GPa but with lower diamond content. At (160±14) GPa, we observe a purely liquid state.”
- Discussion of Figure 4 (line 281 of the manuscript): “Our measurements are expected to achieve thermodynamic equilibrium at the highest temperatures, given the robust diamond formation prior to melting, the exceeding of the glass transition temperature for glassy carbon, and the observed consistency with equilibrium melting models, with that in best agreement, Benedict et al.²⁵, also having similar simulation settings to those used in our DFT-MD calculations.

2. The use of a NN potential is crucial to obtain the results shown in Fig. 3 and for the temperature fitting (NB: DFT-MD with 64 atoms (L=6Å) would lead to an uncertainty of about $2\pi/L \sim 1 \times 10^{-10} \text{ m}^{-1}$ in the determination of the structure factor purely from DFT simulations). The authors must provide more details (as supplementary information) regarding the construction and validation of the NN potential and why they believe it interpolates the full DFT $S(k)$ with the accuracy required to extract temperatures through fits shown in Fig. S3.

To comply with the referee, we now provide additional information as extended data figure No. 5, where we compare $S(k)$ extracted from the NN grid calculation based on 64 atom box simulations, that we used for interpolation as creating the fits, with a simulation with a 216 atom box for 3.7 g/cm and 8000 K. These agree very well, which supports our confidence in the fits with the applied interpolation grid.

3. The use of DFT to determine temperatures is commendable (and DFT is certainly the highest level of theory that can be realistically used to do that). However, while the authors are adamant about the fact the uncertainties introduced by DFT can be controlled only partially, they are less explicit about the implications of such uncertainties on their main claims. For example, it is incorrect to refer to the data in Fig. 4b as "experimental data". The vertical axis is determined by simulations, not by experiments (as instead correctly stated in the main text). Similarly, at page 11, the authors say: "Thus, we assume that the structural representation of liquid carbon in our simulations is correct, which is corroborated by our data." I'm not sure what the authors

mean by this sentence, as T_m is determined by DFT also in the other data reported in Fig. 4b, so Fig. 4b cannot be taken as a validation of the experiments.

The referee is correct that the simulation results are required to extract the temperatures from our X-ray diffraction data (as stressed in the manuscript). We changed the term “experimental data” in the figure caption and a corresponding phrase in the main text (lines 271 and 280 of our manuscript) to “results”.

Regarding the mentioned statement at page 11, we were not referring to the melting temperature but to the fact that our structural data is very well matched by the DFT-MD simulations (with an approximately fourfold coordination), which would not be the case for other models (e.g., the simple Lennard-Jones liquid in the most extreme case). We now clarify this by adding “..., which is corroborated by the excellent match to our diffraction data (e.g., in comparison to the simple Lennard-Jones liquid)” (line 273 of our manuscript). This also provides a better connection to the subsequent sentence stressing the value of our high-quality structure measurements at melting as benchmark for future simulations.

4. The comparison with "simple" liquids is interesting but differences between carbon and a simple liquid are not as surprising as the authors claim. At page 6, for example, they state that coordination in carbon "is in contrast to most other liquids". I would rather say that it is in contrast to simple liquids. There are plenty of other "complex" liquids observed or predicted at those conditions.

We thank the referee for this comment and amended the text according to the suggested formulation “in contrast to simple liquids” (line 156 of the manuscript).

Referee #3 (Remarks to the Author):

This manuscript presents novel results from laser-driven shock compression experiments of glassy carbon samples that were probed by an x-ray free electron laser (XFEL). The experimental methodology was consistent with previous work within the field. The bright XFEL x-rays were used for x-ray diffraction (XRD) measurements to provide structural information of the carbon samples. Complementary velocimetry measurements provided the pressures of the shocked states up to 160 GPa. The XRD data of the high-pressure liquid carbon was compared with density functional theory (DFT) coupled to molecular dynamics (MD) simulations. Overall, the experimental data was of good quality with appropriate uncertainties. Comparison of the data with the DFT-MD simulations were valid, which demonstrated the complex structure of high-pressure liquid carbon. The abstract, introduction, and conclusions were clear, and appropriate references were provided.

We thank the referee for the supportive comments and comply with all suggestions. Details are mentioned below:

To improve the manuscript, the authors should address the following:

1. *Figure 1. The VISAR image should show the scale lengths of time and space axes (e.g. # of ns, and # of um). Similarly, the XRD detectors' images should show their sizes (e.g. # of mm).*

We implemented the requested changes to Fig.1 in our revised manuscript.

Also, the "liquid correlation peaks" are very faint, could the contrast of the images be improved?

The contrast of the depicted raw data images has been improved.

2. *Figure 2. The "blue" line colors for the ambient and 106 GPa states are very similar. Could another color be used for the ambient state (e.g. black)?*

We made the requested change to Fig.2 . The color of the ambient state was changed to grey.

3. *Methods. A brief description of the XRD detectors should be included.*

We added "X-ray diffraction was recorded by two Varex 4343CT flat panel detectors. For more details on the experimental configuration and geometry, see Gorman et al.⁴³" to the Methods section (line 307 of the manuscript).